# Comparison of Zinc, Copper and Selenium Content in Raw, Smoked and Pickled Freshwater Fish

**DOI:** 10.3390/molecules25173771

**Published:** 2020-08-19

**Authors:** Konrad Mielcarek, Anna Puścion-Jakubik, Krystyna J. Gromkowska-Kępka, Jolanta Soroczyńska, Elżbieta Karpińska, Renata Markiewicz-Żukowska, Sylwia K. Naliwajko, Justyna Moskwa, Patryk Nowakowski, Maria H. Borawska, Katarzyna Socha

**Affiliations:** Department of Bromatology, Medical University of Bialystok, ul. Mickiewicza 2D, 15-222 Bialystok, Poland; anna.puscion-jakubik@umb.edu.pl (A.P.-J.); krystyna.gromkowska.kepka@gmail.com (K.J.G.-K.); jolanta.soroczynska@umb.edu.pl (J.S.); elzbieta.karpinska@umb.edu.pl (E.K.); renata.markiewicz@umb.edu.pl (R.M.-Z.); sylwia.naliwajko@umb.edu.pl (S.K.N.); justyna.moskwa@umb.edu.pl (J.M.); patryk.nowakowski@umb.edu.pl (P.N.); bromatos@umb.edu.pl (M.H.B.); katarzyna.socha@umb.edu.pl (K.S.)

**Keywords:** copper, freshwater fish, recommended dietary allowance, selenium, zinc

## Abstract

The aim of the study was to assess the zinc (Zn), copper (Cu) and selenium (Se) content in freshwater fish from Poland. Selected species of raw, smoked and pickled fish were evaluated by atomic absorption spectrometry (AAS). The concentration of Zn, Cu and Se in the examined fish ranged from 1.5 to 49.9 mg/kg, 0.01 to 2.8 mg/kg and 30.9 to 728.2 µg/kg, respectively. One serving of every fish product covered the recommended dietary allowance (RDA) of Zn by 5.38–65.0%, of Cu by 0.42–11.4% and of Se by 12.3–198.6%. A cluster analysis allowed us to distinguish European eel (raw and smoked) based on the Zn content. Additionally, based on the Cu and Se content, pickled common whitefish was differentiated from other species and types of fish products. The discriminant analysis model of smoked fish enabled their classification with a 70% accuracy. Regarding Zn, all forms of the European eel as well as raw and smoked vendace can be considered a source of this element. None of the tested fish can be considered a source of Cu. All products are safe for human consumption with regard to the Zn and Cu content. Almost each form of every species of fish can be considered a source of Se. However, the Se content should be monitored in smoked and pickled common whitefish, pickled bream and pickled vendace.

## 1. Introduction

The World Health Organization recommends regular fish consumption as a measure to prevent lifestyle diseases [1]. Fish are an important component of the human diet since they are a rich source of many essential nutrients. Fish meat is a good source of easily digestible protein containing essential amino acids with a very beneficial composition which facilitate optimal protein utilisation. Fish meat also contains essential unsaturated fatty acids including long-chain polyunsaturated fatty acids from the omega-3 family such as EPA and DHA. These acids exert many positive effects on human health, including the prevention of heart disease and hypertension. They also reduce the mortality of patients with coronary artery disease and have an antiatherosclerotic effect [2]. Fish are a source of mineral elements such as calcium, phosphorous and magnesium [3]. Trace elements of Zn, Cu and Se can also be found in fish tissue [4]. Zn, Cu and Se are micronutrients that are essential for the functioning of the human body and therefore must be included in human diets. On the other hand, they can also be toxic, particularly Se, if absorbed in high concentrations [5,6].

The main role of Zn is based on its function as an integral part of metalloenzymes and as an adjuvant for controlling the activity of specific Zn dependent enzymes [7]. However, high intakes of Zn in relation to Cu can cause a Cu deficiency [8].

Cu is an essential micronutrient which participates in some mechanisms necessary for the functioning of the human body. Cu is a cofactor of enzymes involved in glucose metabolism and hemoglobin production. Cu is not toxic to humans in low concentrations [9], although it may be harmful to cell membranes, DNA and proteins when accumulated in excess [10].

Se is an element with nutritional value. Se can prevent diseases associated with free radical damage including infectious and cardiovascular conditions as well as cancer. In low concentrations, Se is crucial for cellular function and the synthesis of several selenoproteins with antioxidant properties. On the other hand, an excessive intake of Se may result in selenosis [11]. Se enters the food chain from the environment through plants and animals and almost all Se required for selenoprotein synthesis in humans comes from food. Since there is a very small difference between the dose required for proper bodily functions and the toxic dose, it is necessary to know the Se content in food products [12]. Se is also an effective agent for detoxifying mercury (Hg) and protecting against Hg toxicity, whereas selenoproteins are responsible for reducing the toxic effect of Hg [13]. Finding natural food sources of Se is vital, and it has been demonstrated that selenium deficiency affects many inhabitants of Europe, including Poland [14]. 

In accordance with Regulation (EU) No 1169/2011 of the European Parliament and of the Council of 25 October 2011, a product which covers 15% of the recommended dietary allowance (RDA) per portion constitutes a source of a significant amount of minerals [15]. The recommended dietary allowances (RDA) for essential elements for the Polish population are the following: Zn—8 mg/day for women and 11 mg/day for men; Cu—0.9 mg/day; Se—55 μg/day [16,17,18,19].

The tolerable upper intake level (UL) of Zn is 40 mg/day for adults, whereas the lowest observed adverse effect level (LOAEL) of Zn is 60 mg/day. Regarding the safe and toxic doses of Cu, the UL is 10 mg/day. A case report demonstrated that the ingestion of 30 mg/day Cu in the form of tablets for two years, followed by the intake of 60 mg/day Cu for an additional period, resulted in acute liver failure [20]. A study by Rayman et al. [21] showed that a daily intake of 300 μg Se for five years in a country with a moderately low selenium status increased the all-cause mortality ten years later while doses of 100 and 200 μg Se/day resulted in a nonsignificant decrease in mortality during the intervention period, which disappeared after treatment discontinuation.

The Warmia and Mazury Region, known as the Masurian Lake District, is a verdant land dotted with over 2000 lakes. The region was named one of the 28 finalists of the New 7 Wonders of Nature in 2009 [22]. The Warmia and Mazury Region, which is distant from large industrial and urban centers in Poland, is an unpolluted area with low levels of contamination. An analysis of the surface layer of soil performed by the Institute of Soil Science and Plants in Pulawy (Poland) showed that 91.5% of farm-land soils in this region are characterised by a natural content of heavy metals, which is 0°, while 8% of these soils have slightly increased heavy metal contents, that is I°. Only 0.5% of farm-land grounds represented levels II–VI° of heavy metals content (pollution). Additionally, the region has lower than average atmospheric deposition rates of heavy metals [23]. The region specialises in the production of high-quality food, which is connected to the traditionally strong position of agriculture in the area (one of the highest productivity ratios in Poland). The region’s success and its appeal to consumers is based on the dynamic development of local agriculture and traditional food processing methods that utilise regional resources as well as short supply chains [22].

The aim of the study was to examine whether freshwater fish from Warmia and Mazury in Poland are safe for humans in terms of their Zn, Cu and Se content, and whether they can be considered a good source of these elements by assessing their content in edible tissues and calculating the intake of elements in a 150 g portion of raw, smoked and marinated fish.

## 2. Results and Discussion 

### 2.1. Content of Elements in Samples of Fish Products

There were statistically significant differences in the concentrations of the elements studied between different species of fish as well as between groups of raw, smoked and pickled fish, which are presented in Figure 1, Figure 2 and Figure 3.

The Zn concentrations in the investigated freshwater fish products are presented in Figure 1. The total mean concentration of Zn in raw fish samples was 11.1 ± 10.6 mg/kg. Out of all raw samples from the Masurian Lake District, the highest mean Zn content was found in eel samples (32.2 ± 7.6 mg/kg), while the lowest Zn concentration was revealed in common whitefish (3.95 ± 0.9 mg/kg). Szkoda et al. [24] reported similar mean Zn concentrations in raw fish samples from rivers in Poland. Polak-Juszczak and Robak [25] reported lower mean Zn concentrations in raw eel samples (20.7 ± 5.1 mg/kg) from the lakes in northeastern Poland in comparison to the present study. Samples from the Vistula Lagoon and the Vistula River contained 22.5 ± 2.5 mg/kg and 21.5 ± 3.1 mg/kg of Zn, respectively. The lowest Zn content in eels was found in samples from the Szczecin Lagoon (19.4 ± 3.5 mg/kg). Łuczyńska et al. [26] reported lower mean Zn concentrations in the muscles of raw vendace, roach and perch. Marcinkowska and Dobicki [27] investigated samples of raw perch from River Barycza in Poland and the mean Zn concentrations were similar to those obtained in our study (5.2 ± 0.6 mg/kg). Szkoda et al. [24] reported higher mean Zn concentrations in pike-perch samples. Łuczyńska et al. [26] demonstrated a similar Zn content in raw bream and common whitefish from the lakes of the Olsztyn Lake District (North East Poland).

The total mean Zn concentration in smoked fish samples was 13.7 ± 13.1 mg/kg. Out of all the samples of smoked fish, eel (34.6 ± 12.1 mg/kg) had the highest mean Zn concentration and brown trout had the lowest (4.0 ± 1.1 1 mg/kg). Kiczorowska et al. [28] investigated smoked freshwater fish originating from fish farms in the south and east of Poland. The mean Zn content in samples of smoked common whitefish and bream was 8.7 ± 0.17 mg/kg and 3.8 ± 0.3 mg/kg, respectively, and the values were similar to those obtained in the present study. Krełowska-Kułas [29] demonstrated a Zn content similar to the findings of the present study in samples of smoked brown trout from the ponds in Przyborów (Poland).

The total mean Zn concentration in pickled fish samples was 7.3 ± 3.8 mg/kg, with the highest concentration found in eel (13.2 ± 4.3 mg/kg) and the lowest demonstrated in common whitefish (4.6 ± 1.4 mg/kg). There is a lack of data in the literature regarding fish pickled in spirit vinegar in glass jars, which are ready-made products available for sale, and therefore it is difficult to compare data. However, there are reports in the literature regarding fish products marinated in jars by researchers themselves. Cieślik et al. [30] investigated samples of pickled freshwater fish (common carp, rainbow trout) farmed in the Carp Valley (Zator, Poland). The Zn concentrations in samples of the tested pickled freshwater fish species were similar to those of pickled fish from the lakes of Warmia and Mazury.

The concentrations of Cu in freshwater fish products are presented in Figure 2. The total mean Cu concentration in raw fish samples was 0.3 ± 0.2 mg/kg. The highest mean Cu content in raw freshwater fish samples was found in roach (0.5 ± 0.2 mg/kg) and the lowest in pike-perch (0.03 ± 0.1 mg/kg). Research conducted by Kiczorowska et al. [28] in Poland on the edible tissue of raw freshwater fish revealed that the Cu content in tested samples was similar to the values obtained in the present study. Studies by Marcinkowska and Dobicki [27] demonstrated that in meat samples of raw perch from River Barycza in Poland, the mean Cu concentration was 0.79 ± 0.17 mg/kg, which was similar to the value obtained in the present study. Similar Cu concentrations were revealed by Szkoda et al. [24] in their research on raw freshwater fish samples from rivers in Poland. Lidwin-Kaźmierkiewicz et al. [31] studied the Cu in the meat of raw freshwater fish from West Pomerania in Poland and found that the mean muscle concentration of Cu in examined fish ranged from 0.1 to 0.2 mg/kg. Krełowska-Kułas [29] investigated raw freshwater fish from the Vistula River (samples collected in Cracow) and found that the Cu content in tested fish samples ranged from 0.3 to 1.3 mg/kg. The highest Cu concentration was found in the muscle tissue of raw bream. 

Out of all of the samples of smoked fish from the Warmia an Mazury Region, vendace (0.5 ± 0.2 mg/kg) had the highest mean Cu concentration and perch (0.2 ± 0.2 mg/kg) displayed the lowest, while the mean Cu concentration was 0.37 ± 0.2 mg/kg. A study by Kiczorowska et al. [28] found that the Cu concentration in the meat of smoked freshwater fish was similar to that revealed in our study. Cieślik et al. [30] examined samples of smoked freshwater fish farmed in the Carp Valley (Zator, Poland) and demonstrated that the mean Cu content in tested specimens ranged from 0.1 mg/kg to 0.48 mg/kg.

The total mean Cu concentration in pickled fish samples was 0.4 ± 0.4 mg/kg, with the highest concentration found in bream (0.7 ± 0.2 mg/kg) and the lowest concentration demonstrated in pike-perch samples (0.2 ± 0.2 mg/kg). Cieślik et al. [30] investigated samples of pickled freshwater fish from Poland and found that the Cu concentration ranged from 0.1 mg/kg to 0.5 mg/kg in tested specimens. As in the case of Zn, there is a lack of data in the literature regarding the Cu concentration in fish pickled in spirit vinegar in glass jars. 

The concentrations of Se in freshwater fish products are presented in Figure 3. The total mean Se concentration in raw fish samples was 116.8 ± 73.5 μg/kg. The highest mean Se content in raw freshwater fish samples was found in eel (181.3 ± 74.9 μg/kg), while the lowest mean concentration was detected in pike-perch (45.1 ± 13.8 μg/kg). However, a single bream specimen had the highest Se content out of all the raw samples tested (370.1 μg/kg). Polak-Juszczak [32] reported a similar mean Se content in raw freshwater fish from the Vistula Lagoon in Poland. The author investigated species such as perch, roach, bream, eel and pike-perch. The highest mean Se value was found in the meat of eel. 

The total mean Se concentration in smoked freshwater fish samples was 211.5 ± 122.4 μg/kg. The highest mean Se concentration in smoked freshwater fish samples was found in common whitefish (378.6 ± 159.8 μg/kg) and the lowest in perch (113.9 ± 58.1 μg/kg). Cappon [33] determined the Se content in the edible tissue of raw and smoked brown trout caught offshore of Lake Ontario near Rochester, New York. The concentrations of Se in raw and smoked fish samples were lower in comparison to those demonstrated in the present study.

In pickled fish samples, the total mean Se concentration was 376.1 ± 227.7 μg/kg. The highest mean Se concentration was determined in common whitefish (728.2 ± 209.9 μg/kg) and the lowest in roach samples (168.8 ± 99.5 μg/kg). As in the case of Zn and Cu, there is a lack of data in the literature regarding the Se concentration in fish pickled in spirit vinegar in glass jars.

### 2.2. Correlations between Zn, Cu and Se in Edible Parts of Freshwater Fish Products

Several significant correlations were observed between the studied trace elements. Significant positive correlations were found between Se and Zn (*r* = 0.5; *p* < 0.0001) in raw fish samples. Weak positive correlations were also found between Se and Cu (*r* = 0.3; *p* < 0.01) in pickled fish samples. In addition, statistically significant weak negative correlations were observed between Se and Zn (*r* = −0.3; *p* < 0.01) in pickled fish samples. No significant correlations between the examined trace elements were observed in smoked fish samples. In the case of pickled fish, the observed negative correlations may have resulted from the seasoning and marinade used.

Correlations between Cu and Zn (*r* = −0.05), and Cu and Se (*r* = 0.19) were not statistically significant (*p* > 0.05) in the case of raw fish. In the case of smoked fish, the correlation coefficients were the following: (*r* = 0.13) between Cu and Se, (*r* = 0.11) between Cu and Zn, and (*r* = −0.18) between Se and Zn. In the case of pickled fish, an unremarkable correlation existed between the concentrations of Cu and Zn (*r* = −0.04). Benemariya et al. [34] investigated the content of Zn, Cu and Se in fish from a lake in Burundi (Africa). Zn and Cu were positively and highly correlated (*p* < 0.005) while no significant positive correlations were established between the Cu and Se content, or between the Zn and Se content. Sobolev et al. [35], studied fish from Russia. They demonstrated statistically significant correlations between the Se and Cu content, and between the Se and Zn content. Fish may obtain these elements from ambient water through their gills, entire body surface or natural food to ensure normal growth and survival.

### 2.3. Calculated RDA Based on the Consumption of One Serving of Raw, Smoked and Pickled Fish 

The percentages of the RDAs for Zn, Cu and Se met by consuming one serving (150 g) of raw, smoked and pickled fish were calculated and are listed in Table 1. In terms of Zn, a 150 g portion of raw fish provides from 5.4% to 43.9% of the RDA for males and from 7.4% to 60.3% of the RDA for females. One portion of smoked fish covers from 5.5% to 47.3% of the RDA for Zn for males and from 7.6% to 65.0% of the RDA for females. One serving of pickled fish covers from 6.3% to 18.0% of the RDA for Zn for males and from 8.7% to 24.7% for females. Fish meat provides certain amounts of Cu, with one serving of raw fish covering from 0.4% to 7.9%, smoked fish covering from 3.9% to 8.7% and pickled fish covering from 3.3% to 11.4% of the RDA for Cu. The present study revealed that the consumption of one serving of raw fish provides from 12.3% to 49.4% of the RDA for Se for both males and females. The edible tissue of one serving of smoked fish covers from 31.1% to 103.3% of the RDA for Se, while pickled fish provides from 46.1% to 198.6% of the RDA for this element. 

Dietary bioavailability depends on several factors. Cu and Se are comparatively well absorbed, with reported fractional absorption values from mixed diets. Trace elements, such as Zn, are not as well absorbed, with absorption varying widely according to the nutritional status (including body stores) of the individual and composition of their diet [36]. A study by Kiczorowska et al. [28] on raw and smoked freshwater fish from Poland revealed that one serving of fish covered 4.7–8.0% of the RDA for Zn and an average of 6.7% of the RDA for Cu. There is a lack of data in the literature regarding the percentages of the RDAs for Zn, Cu and Se covered by freshwater fish pickled in spirit vinegar in glass jars.

Changes in the trace element content in fish meat can be associated with a number of factors. One of them is cooking. Marimuthu et al. [37] studied snakehead fish (*Channa striatus*, Bloch) cooked using various techniques. The highest Zn content was found in raw fish (5.1 ± 1.0 mg/kg) while the lowest in boiled fish (3.8 ± 2.0 mg/kg). Baked, fried and grilled fish had similar Zn concentrations (4.5 ± 1.0; 4.6 ± 1.0 and 4.1 ± 1.0 mg/kg, respectively). Cieślik et al. [30] studied the impact of processing methods on the content of several elements in fish meat, including Zn. The study was conducted on three species: common carp, rainbow trout and northern pike. The results were inconclusive. The authors demonstrated that the smoking process caused an increase in the Zn content in common carp (from 8.3 ± 0.3 to 11.5 ± 1.0 mg/kg) and rainbow trout meat (from 4.5 ± 0.04 to 7.9 ± 0.7 mg/kg), but a decrease in northern pike meat (from 6.2 ± 0.01 to 4.43 ± 0.02 mg/kg). Marinating, on the other hand, reduced the Zn content in marinated common carp (5.5 ± 0.02 mg/kg), but increased the content in rainbow trout (6.2 ± 0.01 mg/kg). 

The bioaccumulation of micronutrients in fish is influenced by a number of factors including the age of the fish, their length and weight [38]. Zn accumulates in fish tissue to a small degree since it is retained and deposited in the gills. This is explained by the fact that Zn is absorbed into the bloodstream more easily, compared to other elements such as cadmium or nickel. Its bioaccumulation depends on the affinity for erythrocyte membranes. In addition, Zn displays low toxicity to freshwater fish. Zn content in fish meat is also affected by the composition of the feed and interactions between the elements present in food [39]. Different concentrations of Se in studied fish species may be caused by, among other reasons, leaching it from the environment by groundwater, variable parameters of the aquatic environment, the capacity of migration and food habits [40]. In addition, differences in element contents may result from different fishing locations and timings, and the geochemical properties of the soil or feed type [41]. Pickled fish products also contain vegetables and seasoning that can increase the content of elements in the final product. The Chief Inspectorate of Environmental Protection in Poland monitors lake water quality standards. Assessments of the condition of lakes in years 2010–2018 revealed that the concentrations of Zn, Cu and Se were above the limit of quantification in lakes from which the studied fish were caught [42,43]. This can be the main reason for the observed element content in studied fish meat.

### 2.4. Elements of Chemometrics

**Cluster analysis.** Dendrograms in Figure 4a–c illustrate the results of the cluster analyses for the individual elements studied (Zn—Figure 4a, Cu—Figure 4b, Se—Figure 4c). This method is used to search for specific subsets within a group of objects related to each other in terms of a parameter, differing from other clusters. Differential variables were included in the analysis: the Zn, Cu and Se concentration. Agglomeration using the Ward method was used, and the distance was measured by the Euclidean distance. A cluster analysis of Zn primarily distinguished one main cluster, classifying raw and smoked European eel and other fish species and types: raw, smoked and pickled (Figure 4a). An analysis of Cu showed differences in the concentration of this element between pickled common whitefish, and other fish species and genera (Figure 4b). The data regarding the Se concentration indicated two main groups—the first contained pickled common whitefish and the second contained the remaining species and types of fish (Figure 4c). In accordance with the principles of cluster analysis, the fish sample data were taken randomly so that the obtained results could be generalised.

**Discriminant analysis.** Discriminant analysis is a statistical technique which shows variables that discriminate between groups. It allows for the studying of differences between groups, taking into account several variables. The purpose was to search for rules that assign multidimensional objects to a group with the fewest possible classification errors. In the case of smoked and pickled fish, all three tested elements were indicated as significant variables in the model: the content of Cu, Se and Zn. The discriminant analysis of fresh fish indicated that the concentrations of Zn and Cu were significant. Lambda indicated that Zn makes the biggest contribution to the overall discrimination in the case of fresh and smoked fish while, in the case of pickled fish, Zn and Se are the key contributors. Figure 5, Figure 6 and Figure 7 show scatter plots for the obtained data on raw fish (Figure 5), smoked fish (Figure 6) and pickled fish (Figure 7). The model created for raw fish classified the studied fish species with a 47% accuracy. As shown in the figures, the first discriminant function primarily distinguishes European eels and the second function distinguishes pike-perch (Figure 5A–C). The model created for smoked fish classified the studied cases with a 70% accuracy. The first discriminant function distinguished primarily European eel and vendace. Common whitefish was discriminated from other species by the second function (Figure 6A–C). The model for pickled fish classified studied cases with 56% accuracy. Function 1 primarily distinguished common whitefish only (Figure 7A–C). Therefore, only the model for smoked fish can be considered satisfactory. In the case of raw and pickled fish, other factors should be sought, allowing for their classification with an accuracy higher than 70%. 

## 3. Materials and Methods

### 3.1. Study Areas and Sample Collection

Samples of selected species of freshwater fish (*n* = 212) were investigated. There were 74 samples of raw fish, 66 samples of smoked fish and 72 samples of fish pickled in spirit vinegar (not all investigated fish species were available in all three forms: raw, smoked and pickled). Pickled fish products contained fish which were first boiled, smoked, baked or fried and then pickled in a concoction of spirit vinegar, salt, sugar, food gelatin, vegetables and seasoning. Species such as brown trout (*Salmo trutta* morpha *lacustris* L., smoked: *n* = 10), common bream (*Abramis brama* L., raw: *n* = 14, smoked: *n* = 10, pickled: *n* = 10), common roach (*Rutilus rutilus* L., raw: *n* = 10, pickled: *n* = 12), common perch (*Perca fluviatilis* L., raw: *n* = 10, smoked: *n* = 10, pickled: *n* = 10), common whitefish (*Coregonus lavaretus* L., raw: *n* = 10, smoked: *n* = 12, pickled: *n* = 10), European eel (*Anguilla anguilla* L., raw: *n* = 10, smoked: *n* = 13, pickled: *n* = 10), pike-perch (*Sander lucioperca* L., raw: *n* = 10, pickled: *n* = 10) and vendace (*Coregonus albula* L., raw: *n* = 10, smoked: *n* = 11, pickled: *n* = 10) were collected from fishing farms in Warmia and Mazury Region in Poland on Roś, Nidzkie, Ukiel, Czos, Śniardwy and Niegocin lakes (Figure 8) in the towns and cities of Mikołajki, Mrągowo, Olsztyn, Pisz, Piękna Góra, Ruciane Nida and Orzysz. Samples of raw, hot smoked and pickled fish were collected (purchased from fishing farm shops) in 2017 and 2018. 

### 3.2. Reagents and Solutions

Ultrapure water and analytical grade chemicals were used throughout the procedures. A system of water purification, Simplicity 185 (Millipore, Bedford, MA, USA), was used to provide the ultrapure water necessary for all tests. Stock solutions of 1000 mg/L (Merck, Darmstadt, Germany) were used for the preparation of standard solutions of Zn, Cu and Se. Spectrally concentrated ultrapure nitric acid 69% (69% HNO_3,_ Tracepur, Merck, Darmstadt, Germany) was used in the acid digestion step. For Se determination, the matrix modifier was used (1500 mg/L of palladium(II) nitrate and 900 mg/L of magnesium nitrate hexahydrate), freshly prepared for each analysis (Merck, Darmstadt, Germany).

### 3.3. Fish Samples Preparation Procedure

Firstly, the muscle tissues of fish samples were separated from inedible parts. In the case of pickled fish, the jelly was separated from muscles tissues. Then, samples were homogenised using a mechanical homogeniser (OMNI International TH-02, OMNI International, Kennesaw, GA, USA). Finally, the samples (average 1.2 g) were mineralised in a 69% HNO_3_ (4.0 mL) using a closed microwave digestion system (Berghof, Speedwave, Germany). The volume of samples after digestion ranged from 5.0 to 6.0 mL.

### 3.4. Determination of Zn, Cu and Se in Fish Samples

The concentration of Zn was determined by atomic absorption spectrometry (AAS) (Z-2000 Tandem Flame/Furnace AA Spectrophotometer, Hitachi, Japan) with flame atomisation in an acetylene—air flame at a wavelength of 213.9 nm with a Zeeman background correction. The concentrations of Cu and Se were determined by AAS with electrothermal atomisation at the wavelength of 196.0 nm and 224.8 nm, respectively, with a Zeeman background correction. In the case of Se, a palladium-magnesium matrix modifier was used.

### 3.5. Quality Control

Quality control was performed using certified reference materials: MODAS-5 Cod Tissue and MODAS-3 Herring Tissue. The Department of Bromatology of the Medical University of Bialystok participated in the quality control programme for the trace element analysis in cod and herring tissues supervised by the Institute of Nuclear Chemistry and Technology, Poland [44]. The results of the quality control analyses corresponded with the reference values. The accuracy of the method was 0.29%, 0.93% and 1.48% and the coefficient of variation was 1.25%, 2.40% and 4.59% for Zn, Cu and Se, respectively. The detection limit of the method was 0.014 mg/kg, 0.0004 mg/kg and 1.65 μg/kg for Zn, Cu and Se, respectively.

### 3.6. Statistical Analysis

The statistical analyses were performed using Statistica v.13.0 software. The assessment of normality was conducted using Kolmogorov-Smirnov tests. The differences between independent groups were examined using a Student’s t-test. The correlations were calculated and tested using Pearson’s test. A p value of <0.05 was considered statistically significant. In addition, selected chemometry analyses were performed to classify fish, i.e., a cluster and discriminant analysis. To perform cluster analyses, the data were standardised. In the case of discriminant analysis, the model was calibrated in order to increase the discriminant power—it allowed for the selection of variables that had the highest share in discrimination. The percentages of RDAs for elements were calculated in reference to the consumption of one 150 g portion of fish, which is the accepted average fillet weight [45]. 

## 4. Conclusions

All the processed fish had significantly higher Se concentrations than raw fish. Almost each form (raw, smoked and pickled) of every species of freshwater fish can be considered a source of Se and should be included in people’s diets. Due to the high concentration of Se in smoked and pickled common whitefish, pickled common bream and pickled vendace, the content of Se should be monitored in these products and their elimination from diets should be considered by individuals with a high intake of Se from other sources. Raw and smoked fish had significantly higher Zn concentrations than pickled fish. It is also worth noting that all forms of the European eel, as well as raw and smoked vendace, are a source of Zn. All forms of freshwater fish are safe for human consumption in regard to Zn and Cu content. None of the studied raw, smoked and pickled freshwater fish species are sources of Cu. A cluster analysis and discriminant analysis can be helpful in differentiating and classifying the selected species of fish and fish products based on the content of Se, Zn and Cu.

## Figures and Tables

**Figure 1 molecules-25-03771-f001:**
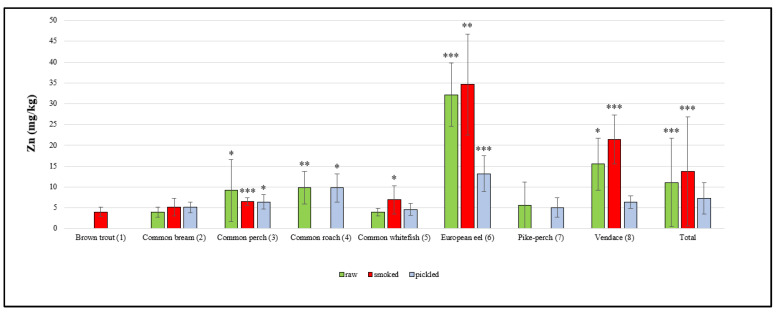
Zn concentration in raw, smoked and pickled freshwater fish product samples (mg/kg). Statistical analysis showed significant differences in the Zn content between species in each group of freshwater fish products (* *p* < 0.05; ** *p* < 0.01; *** *p* < 0.001); raw: *p* < 0.05, (3 vs. 2, 5) and (8 vs. 2, 3, 4, 5, 7); *p* < 0.01, (4 vs. 2, 5); *p* < 0.001, (6 vs. 2, 3, 4, 5, 7, 8); smoked: *p* < 0.05, (5 vs. 1); *p* < 0.01, (6 vs. 1, 2, 3, 4, 5, 7, 8); *p* < 0.001, (3 vs. 1) and (8 vs. 1, 2, 3, 4, 5, 7); pickled: *p* < 0.05, (3 vs. 5) and (4 vs. 2, 3, 5, 7, 8); *p* < 0.001, (6 vs. 2, 3, 5, 7, 8); total: *p* < 0.001, (smoked vs. pickled) and (raw vs. pickled).

**Figure 2 molecules-25-03771-f002:**
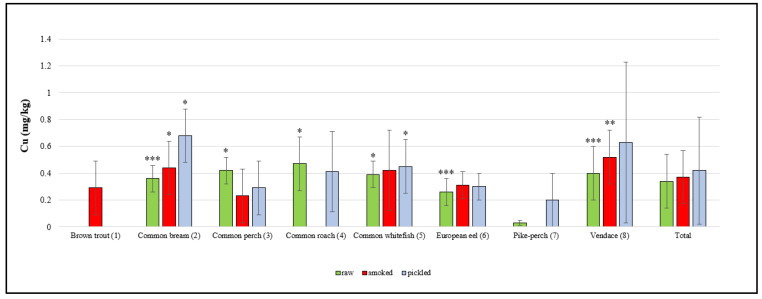
Cu concentration in raw, smoked and pickled freshwater fish product samples (mg/kg). Statistical analysis showed significant differences in the Cu content between species in each group of freshwater fish products (* *p* < 0.05; ** *p* < 0.01; *** *p* < 0.001); raw: *p* < 0.05, (3 vs. 6, 7), (4 vs. 6, 7) and (5 vs. 6, 7); *p* < 0.001, (2 vs. 7), (6 vs. 7) and (8 vs. 7); smoked: *p* < 0.05, (2 vs. 3); *p* < 0.01, (8 vs. 1, 3, 6); pickled: *p* < 0.05, (2 vs. 1, 3, 4, 5, 6, 7) and (5 vs. 6, 3, 7).

**Figure 3 molecules-25-03771-f003:**
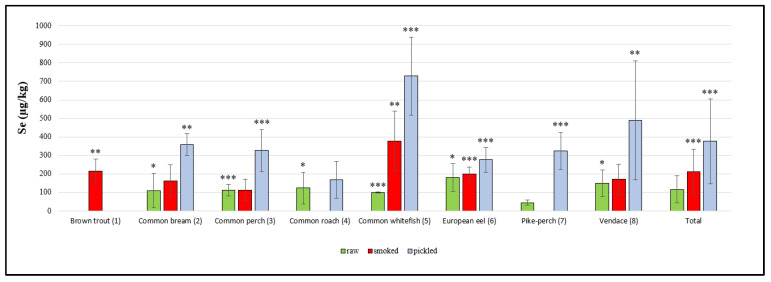
Se concentration in raw, smoked and pickled freshwater fish product samples (µg/kg). Statistical analysis showed significant differences in the Se content between species in each group of freshwater fish products (* *p* < 0.05; ** *p* < 0.01; *** *p* < 0.001); raw: *p* < 0.05, (2 vs. 7), (4 vs. 7), (6 vs. 3, 5, 7) and (8 vs. 5, 7); *p* < 0.001, (3 vs. 7) and (5 vs. 7); smoked: *p* < 0.01, (1 vs. 3) and (5 vs. 1, 2, 3, 4, 6, 7, 8); *p* < 0.001 (6 vs. 3); pickled: *p* < 0.01, (2 vs. 6, 4) and (8 vs. 4); *p* < 0.001, (3 vs. 4), (5 vs. 1, 2, 3, 4, 6, 7), (6 vs. 4) and (7 vs. 4); total: *p* < 0.001, (smoked vs. raw), (pickled vs. raw) and (pickled vs. smoked).

**Figure 4 molecules-25-03771-f004:**
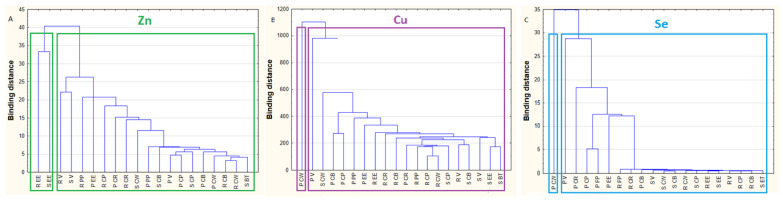
Cluster analysis of raw, smoked and pickled freshwater fish in terms of their Zn content (**A**), Cu content (**B**) and Se content (**C**). P CB—pickled common bream, P CP—pickled common perch, P CR—pickled common roach, P CW—pickled common whitefish, P EE—pickled European eel, P PP—pickled pike-perch, P V—pickled vendace, R CB—raw common bream, R CP—raw common perch, R CR—raw common roach, R CW—raw common whitefish, R EE—raw European eel, R PP—raw pike-perch, R V—raw vendace, S BT—smoked brown trout, S CB—smoked common bream, S CP—smoked common perch, S CW—smoked common whitefish, S EE—smoked European eel, S V—smoked vendace.

**Figure 5 molecules-25-03771-f005:**
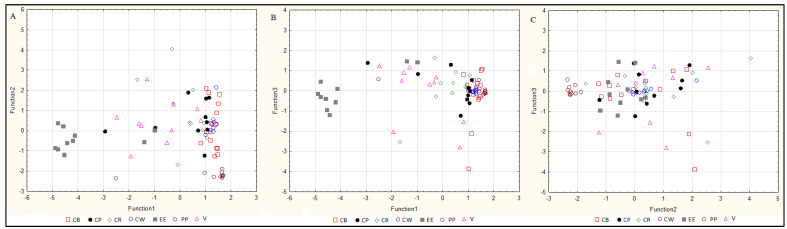
Discriminant analysis of raw freshwater fish. CB—common bream, CP—common perch, CW—common whitefish, EE—European eel, PP—pike-perch, V—vendace. (**A**). Function 1 vs. function 2, (**B**). Function 1 vs. function 3, (**C**). Function 2 vs. function 3.

**Figure 6 molecules-25-03771-f006:**
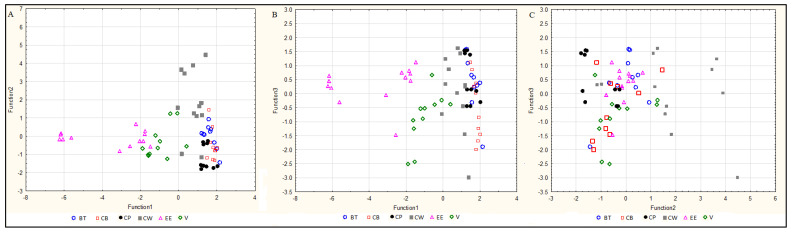
Discriminant analysis of smoked freshwater fish. BT—brown trout, CB—common bream, CP—common perch, CW—common whitefish, EE—European eel, V—vendace. (**A**). Function 1 vs. function 2, (**B**). Function 1 vs. function 3, (**C**). Function 2 vs. function 3.

**Figure 7 molecules-25-03771-f007:**
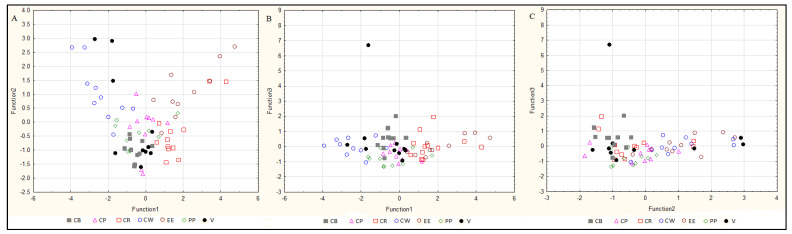
Discriminant analysis of pickled freshwater fish. CB—common bream, CP—common perch, CR—common roach, CW—common whitefish, EE—European eel, PP—pike-perch, V—vendace. (**A**). Function 1 vs. function 2, (**B**). Function 1 vs. function 3, (**C**). Function 2 vs. function 3.

**Figure 8 molecules-25-03771-f008:**
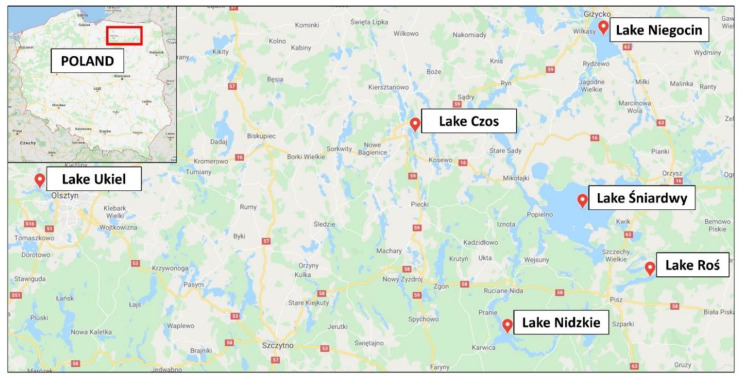
Location of the study area—lakes of Warmia and Mazury Region, Poland.

**Table 1 molecules-25-03771-t001:** Recommended dietary allowance (%) of Se, Cu and Zn after the consumption of 150 g of fish product.

Fish Species	RDA (%) For Se			RDA (%) For Cu			RDA (%) For Zn					
	Raw	Smoked	Pickled	Raw	Smoked	Pickled	Raw		Smoked		Pickled	
							Male	Female	Male	Female	Male	Female
**Brown trout** ***(Salmo trutta*** **morpha** ***lacustris)***	NA	59.1%	NA	NA	4.8%	NA	NA	NA	5.5%	7.6%	NA	NA
**Common bream** *(Abramis brama)*	30.1%	44.7%	97.9%	6.0%	7.3%	11.4%	5.4%	7.5%	7.0%	9.6%	7.0%	9.6%
**Common perch** (*Perca fluviatilis)*	30.5%	31.1%	89.1%	7.0%	3.9%	4.8%	12.5%	17.2%	9.0%	12.3%	8.8%	12.1%
**Common roach** (*Rutilus rutilu*)	33.7%	NA	46.1%	7.9%	NA	6.8%	13.5%	18.5%	NA	NA	13.4%	18.4%
**Common whitefish** *(Coregonus lavaretus)*	26.8%	103.3%	198.6%	6.5%	7.0%	7.5%	5.4%	7.4%	9.5%	13.0%	6.3%	8.7%
**European eel** *(Anguilla anguilla)*	49.4%	54.4%	75.4%	4.4%	5.1%	5.0%	43.9%	60.3%	47.3%	65.0%	18.0%	24.7%
**Pike-perch** (*Sander lucioperca*)	12.3%	NA	88.7%	0.4%	NA	3.3%	9.1%	12.5%	NA	NA	6.9%	9.5%
**Vendace** (*Coregonus albula*)	40.9%	46.6%	133.4%	6.7%	8.7%	10.5%	21.1%	29.1%	29.3%	40.2%	8.7%	11.9%
**RDA ^a^**	55 µg/day			0.9 mg/day			Male: 11 mg/day, Female: 8 mg/day					

^a^ Food Standards for Polish Population [19]; NA—not analysed.

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
