# Peer review of "Comparison of Zinc, Copper and Selenium Content in Raw, Smoked and Pickled Freshwater Fish"

_molecules, 2020, doi:10.3390/molecules25173771_

Round 1
Reviewer 1 Report
The manuscript, although not very innovative, could be of interest for food science. However for its publication some minor revisions are recommended.
The Authors showed several correlations between Zn, Cu and Se in edible parts of freshwater fish products. I would like to read a broader discussion related on these correlations.
Legends to all figures must be completely rewritten. They must be more self-explanatory.
Paragraph numbers are incorrect and need to be revised.
The whole manuscript must be revised in its form.
Author Response
Point 1: The manuscript, although not very innovative, could be of interest for food science. However for its publication some minor revisions are recommended.
Response 1: We sincerely thank the Reviewer for suggestions and valuable comments, which were of great help in revising the manuscript and improved our paper. We appreciate the Reviewer's feedback.
Point 2: The Authors showed several correlations between Zn, Cu and Se in edible parts of freshwater fish products. I would like to read a broader discussion related on these correlations.
Response 2: According to suggestion, we added a broader discussion related to these correlations. To expand this topic, we also include other correlations, likewise negatives. We found studies, which authors included correlations between Zn, Cu and Se in meat tissues of freshwater fish. We also described the likely causes of the correlations present:
Several significant correlations were observed between studied trace elements. Significant positive correlations were found between Se and Zn (r = 0.5; p<0.0001) in raw fish samples. Weak positive correlations were also found between Se and Cu (r = 0.3; p<0.01) in pickled fish samples. In addition, statistically significant weak negative correlations were observed between Se and Zn (r = - 0.3; p<0.01) in pickled fish samples. No significant correlations between the examined trace elements were observed in smoked fish samples. In the case of pickled fish, the observed negative correlations may have resulted from the seasoning and marinade used.
Correlations between Cu and Zn (r = -0.05), and Cu and Se (r = 0.19) were not statistically significant (p> 0.05) in the case of raw fish. In the case of smoked fish, the correlation coefficients were the following: (r = 0.13) between Cu and Se, (r = 0.11) between Cu and Zn, and (r = -0.18)
between Se and Zn. In the case of pickled fish, an unremarkable correlation existed between the concentrations of Cu and Zn (r = -0.04). Benemariya et al. [34] investigated the content of Zn, Cu and Se in fish from a lake in Burundi (Africa). Zn and Cu were positively and highly correlated (p < 0.005) while no significant positive correlations were established between Cu and Se, and between Zn and Se. Sobolev et al. [35], studied fish from Russia. They demonstrated statistically significant correlations between Se and Cu, and between Se and Zn. Fish may obtain these elements from ambient water through gills, entire body surface or natural food to ensure normal growth and survival.
Point 3: Legends to all figures must be completely rewritten. They must be more self-explanatory.
Response 3: We are grateful to the Reviewer for pointing this out. We have made improvements in legends to figures, as well as in the figures themselves. We have also made corrections in figure descriptions. In our opinion, now the figures are more self-explanatory.
Point 4: Paragraph numbers are incorrect and need to be revised.
Response 4: We kindly apologize for this. We think it was the MS Word program error. We have improved the numbers of every paragraph.
Point 5: The whole manuscript must be revised in its form.
Response 5: We have put everything in the right order. We have adapted everything to the requirements of the journal. Additionally, we have subjected the publication to professional English editing services.
Reviewer 2 Report
Review of manuscript molecules-878573 “Comparison of zinc, copper and selenium content in raw, smoked and pickled freshwater fish with chemometrics analysis”
Summary
In this manuscript, the authors present the comparison of the content of three metals (zinc, copper and selenium) in fishes from a region in Northern Poland. Besides, the authors evaluate these differences in the metal content associated with different ways of preparing the fish dishes.
Commentaries
In my opinion, this manuscript is mainly well designed and written, but some modifications should be addressed before warranting its publication.
- Overall, an English-language revision of the text would benefit the quality of the manuscript. The English writing could be improved both considering style and grammar issues.
- Consider a revision of the title. After reading the manuscript, I do not feel that the performed chemometric analysis are relevant enough to appear in the title.
- Abstract. Consider rewriting it to highlight the main results obtained in the study and present the most interesting conclusions derived from these results.
- Last sentence of the abstract. Something should be said regarding the safety of selenium.
- Introduction. Details regarding the safe/toxic concentrations for the different metals should be given
- Introduction. Paragraph lines 61-72. I think that the description of the region should be focused on the scientific-relevant details. More touristic aspects should be removed.
- Results. Please, use significant numbers correctly. Follow guidelines here: https://doi.org/10.1016/j.aca.2015.01.017
- Results. Are some of the reported copper concentrations below the LOD (0.42 mg/kg as stated in the methods section)? Please, revise.
- Results – Correlation study. I think that this study should be enlarged, also presenting the values of the non-significant correlations. In addition, although being significant, the R values are rather low. Can the authors give some explanation of this fact?
- Table 1. Is the footnote necessary? No max, min, sd are defined in the table.
- Results – Chemometrics. All this section is rather weak. The part of the hierarchical clustering is acceptable, but the section of the discriminant analysis should be removed (or redone/rewritten). The authors do no explain what is seen in the figures and, also, the obtained models cannot be considered satisfactory with classifications under 50-70%.
- Methods. More details regarding the application of the chemometrics methods should be given. Data pretreatment? How was the calibration/validation performed?
Round 2
Reviewer 2 Report
After the evaluation of the changes performed by the authors in the revised version, I consider that the major weak points of the manuscript have been amended. So, I think that the manuscript can be published in its present form.
Regarding the question raised by the authors, I consider that the third tittle is the best option “Comparison of zinc, copper and selenium content in raw, smoked and pickled freshwater fish”